# TCF-1 Is Required for CD4 T Cell Persistence Functions during AlloImmunity

**DOI:** 10.3390/ijms24054326

**Published:** 2023-02-21

**Authors:** Mahinbanu Mammadli, Liye Suo, Jyoti Misra Sen, Mobin Karimi

**Affiliations:** 1Department of Microbiology and Immunology, SUNY Upstate Medical University, Syracuse, NY 13210, USA; 2Department of Pathology, SUNY Upstate Medical University, Syracuse, NY 13210, USA; 3National Institute on Aging-National Institute of Health, 251 Bayview Boulevard, Baltimore, MD 21224, USA; 4Center of Aging and Immune Remodeling and Immunology Program, Department of Medicine, Johns Hopkins School of Medicine, Baltimore, MD 21224, USA

**Keywords:** TCF-1, CD4 T cells stemness, CD4 T cell serum level cytokine production, alloimmunity

## Abstract

The transcription factor T cell factor-1 (TCF-1) is encoded by Tcf7 and plays a significant role in regulating immune responses to cancer and pathogens. TCF-1 plays a central role in CD4 T cell development; however, the biological function of TCF-1 on mature peripheral CD4 T cell-mediated alloimmunity is currently unknown. This report reveals that TCF-1 is critical for mature CD4 T cell stemness and their persistence functions. Our data show that mature CD4 T cells from TCF-1 cKO mice did not cause graft versus host disease (GvHD) during allogeneic CD4 T cell transplantation, and donor CD4 T cells did not cause GvHD damage to target organs. For the first time, we showed that TCF-1 regulates CD4 T cell stemness by regulating CD28 expression, which is required for CD4 stemness. Our data showed that TCF-1 regulates CD4 effector and central memory formation. For the first time, we provide evidence that TCF-1 differentially regulates key chemokine and cytokine receptors critical for CD4 T cell migration and inflammation during alloimmunity. Our transcriptomic data uncovered that TCF-1 regulates critical pathways during normal state and alloimmunity. Knowledge acquired from these discoveries will enable us to develop a target-specific approach for treating CD4 T cell-mediated diseases.

## 1. Introduction

T cell factor-1 (TCF-1, encoded by *Tcf7*) regulates T cell development, cell fate specification, and maintenance of tissue homeostasis [1]. TCF-1 plays a critical role in T cell responses to viral infection, cancer, and autoimmunity [1,2,3,4]. TCF-1 is a key mediator of both Th1 and Th17 cytokines, and T-bet expression has been shown to be regulated by TCF-1 [5,6]. Studies from germline TCF-1 knock-out mice and enforced expression models have demonstrated that T-bet recruits the transcriptional repressor Bcl6 to the TCF-1 promoter and inhibits TCF-1 gene expression [7]. These data suggest that T-bet can regulate the role of TCF-1 in Th1 differentiation or effector function. A major limitation of these studies was the use of TCF-1 germline knock-out mice or Wnt pathway inhibitors that are not specific. CD4 T cells with a higher TCF-1 expression have been shown to have self-renewing capacity, while CD4 T cells with a lower TCF-1 expression do not, at least in the context of a viral infection [8]. Studies have also shown that both LEF-1 and TCF-1 play a central role in Th2 cell development by suppressing Th2 specific cytokines and the induction of GATA-3 [9]. TCF-1 has been shown to play a critical role in Tfh cells by regulating IL-4/STAT-6 signaling to control the differentiation of CD4 cytolytic cells [9,10,11,12]. TCF-1 has also been shown to inhibit IL-17 during the early stages of development, limiting peripheral Th17 cells [13]. However, to our knowledge, TCF-1 has not been studied in the context of alloimmunity, which is a different process from canonical T cell activation.

To study the role of TCF-1 in CD4 T cells in a clinically relevant model, we utilized an allogeneic hematopoietic stem cell transplantation (allo-HSCT) murine model. Allo-HSCT is a curative option in the treatment of aggressive malignant and non-malignant blood disorders [14,15]. However, the benefits of allo-HSCT can be compromised by graft-versus-host disease (GvHD), a prevalent and morbid complication of allo-HSCT. We used a unique mouse strain that has a deletion of TCF-1 in mature T cells, rather than a global deletion [16,17]. The progeny of a TCF-1 flox/flox mouse was crossbred with a CD4 cre+/+ C57BL/6 mouse experiencing a deletion of TCF-1 in all T cells at the double-positive (DP) phase of development when all T cells express CD4 [18]. This approach allowed us to directly investigate the role of TCF-1 on peripheral CD4 T cells [19]. Our compelling data demonstrate that the loss of TCF-1 in CD4 T cells reduces both the severity and the persistence of GvHD, leading to improved survival of recipient mice following transplantation. We also showed that TCF-1 significantly impacts CD4 T cell activation memory formation. Our data also uncovered that TCF-1 differentially impacts chemokine expression, however, donor CD4 T cells from TCF-1 cKO mice had no impact on donor T cell migration to GvHD target tissues. Several lines of evidence have suggested that CD28 is critical for the CD4 T cell persistence function [20,21]. Our data showed that CD4 T cells from TCF-1 cKO mice expresses significantly less CD28 co-stimulatory molecules. For the first time, we provide evidence that TCF-1 regulates CD28 expression in T cells [17]. The loss of CD28 in CD4 T cells has no impact on CD4 T cell initial cytokine production [22]. We uncovered that mature CD4 T cells from TCF-1 cKO mice produce significantly less Th1 cytokines but show increased production of Th2 cytokines in the in-vivo alloimmune disease model, which was previously unknown. Th1 cytokines have been shown to cause donor T cell proliferation/expansion and to significantly amplify the development of GvHD (more than Th2 cytokines) [23,24,25,26]. Our transcriptomic data uncovered that pre- and post-transplanted CD4 T cells from TCF-1 cKO mice have an altered expression of pathways related to apoptosis and cell death, T cell-mediated processes, cytokine production, and cell adhesion. Overall, our data demonstrate that TCF-1 significantly impacts peripheral CD4 T cell phenotype, cytokine production, chemokine expression, and cell survival, and alters the genetic profile in a clinically relevant murine model. Thus, our findings contribute significantly to understanding the role of TCF-1 in CD4 T cells in alloimmunity.

## 2. Results

### 2.1. TCF-1 Regulates CD4 T Cell Phenotype and Memory Formation

Previous research on TCF-1 focused on T cell development and canonical activation [2,27]. These studies primarily used a global TCF-1 knockout, resulting in limited production of mature T cells. We sought to examine the role of TCF-1 in mature T cells, so to overcome this limitation we employed mice with a T-cell-specific deletion of TCF-1 using TCF-1 flox/flox mice bred with CD4 cre+/+ mice [28,29]. This mouse strain has TCF-1 deleted in all CD4 and CD8 T cells at the DP phase, allowing production of mature T cells with a TCF-1 deletion. The CD4 T cell phenotype has been extensively studied in response to viral infections, cancer, and GvHD [30,31,32]. More specifically, CD4 T cell phenotype plays a central role in autoimmune disorders [32,33]. However, whether TCF-1 regulates CD4 T cell activation, effector, and central memory phenotypes in peripheral CD4 T cells is unclear.

To investigate whether TCF-1 regulates these critical functions of CD4 T cells, we examined naïve CD4 T cells from either WT C57Bl/6, CD4 cre+/+, or TCF-1 cKO mice. CD4 T cells were isolated from spleens, and we confirmed by flow cytometry the loss of TCF-1 in CD4 T cells (Figure 1A). Next, we examined the expression of activation markers, such as CD44 and CD122 on CD4 T cells. CD4 T cells expressing these markers have been shown to induce significantly less or no GvHD [32,34]. Our data showed that splenic naïve CD4 T cells from TCF-1 cKO mice expressed significantly more CD44 and CD122 markers, compared to CD4 T cells from control mice including WT and CD4 cre+/+ strains (Figure 1B,C).

These data uncovered that TCF-1 plays a suppressive role in CD4 T cell activation and the loss of TCF-1 affects the CD4 T cell phenotype [17,35]. We also examined whether CD4 T cells from TCF-1 cKO mice might impact the T-box transcription factor family members Eomes and T-bet [36], which are important in anti-tumor responses of T cells and play central roles in T cell mediated GvHD and alloimmunity [37]. The key transcription factor T-bet is known to regulate the balance between the effector and central memory phenotype of T cells [38,39]. Our data showed that CD4 T cells from TCF-1 cKO mice expressed significantly more T-bet than CD4 T cells from control WT, and CD4 cre+/+ mice (Figure 1D). The data with higher T-bet expressions in TCF-1 cKO mice correlate with higher expressions of CD122 and CD44 in CD4 T cells from TCF-1 cKO mice [40,41,42]. However, we did not observe any differences in Eomes expression in splenic naïve CD4 T cells from WT, compared to TCF-1 cKO mice (Appendix A). 

Since both effector (EM) and central (CM) memory cells have been shown to play a significant role in CD4 T cell-mediated diseases, including GvHD [32], we wanted to examine the memory phenotype in CD4 T cells from TCF-1 cKO mice. Effector memory cells were defined as CD44^high^ CD62L^low^, central memory cells defined as CD44^high^ CD62L^high^, and naïve CD4 T cells defined as CD44 ^low^ CD62L^high^ subgroups. Surprisingly, our data showed that there was a significant increase in both EM and CM cells and significantly fewer naïve cells in TCF-1 cKO mice, compared to WT, and CD4 cre+/+ mice (Figure 1E). These findings shows that TCF-1 does play a significant role in the EM and CM formation in CD4 T cells [17,43,44]. 

Because ICOS has been shown to play a critical role in effector memory, CD4 T cell survival, and maintenance [45,46,47], we wanted to examine the expression of ICOS in the effector memory of CD4 T cells from TCF-1-deficient and WT, or control and CD4 cre+/+ mice. Our data uncovered that the splenic EM memory of CD4 T cells from TCF-1 cKO mice express ICOS significantly less frequently, compared to CD4 T cells from other control mice, suggesting that even though TCF-1-deficient mice have more effector memory cells, their survival and maintenance may be negatively affected, providing evidence that TCF-1 is required for CD4 T cell persistence functions (Figure 1F). 

Published data have shown that CXCR3 expression is directly linked to T-bet expression in EM cells during viral infections, and that TCF-1 controls the chemokine receptor expression in CD4 T cells and their trafficking to GvHD target organs [48,49]. A molecular analysis also showed that TCF-1 has been implicated in T cell migration [50,51]. Our data showed that naïve splenic CD4 T cells from TCF-1 cKO mice express CXCR3 significantly more frequently, which suggests that TCF-1 CD4 T cells might migrate to GvHD target organs more easily than CD4 T cells from WT or CD4cre+/+ control mice (Figure 1G).

The differences we observed in the phenotype of the CD4 T cells between WT and TCF-1 cKO mice could be cell-intrinsic (due directly to gene deletion within the cell) or cell-extrinsic (due to different microenvironments caused by gene deletion) [52,53]. To determine whether the above-mentioned differences of CD4 T cell phenotypes from TCF-1 cKO mice are cell-intrinsic or cell-extrinsic, we mixed _TCD_BM from WT mice with the congeneric marker CD45.1 along with bone marrow from TCF-1 cKO mice with the congeneric marker CD45.2 at a 1:4 ratio. These ratios were determine based on our previous publication [54]. 

To determine whether the phenotypic effects we observed were cell-intrinsic or cell-extrinsic, we developed a chimeric mouse model. Briefly, we mixed bone marrow from WT and TCF-1 cKO mice at a 1:4 (WT:TCF) ratio for a total of 50 × 10^6^ BM cells, then used this mixture to reconstitute lethally irradiated into mice with the congeneric marker Thy.1. to generate a mixed chimera model. Nine to 10 weeks post transplantation, we bled the recipient Thy1.1 mice to confirm the successful reconstitution and creation of a chimera model [17]. Ten weeks post-chimerization, we euthanized recipient Thy1.1 mice and used a flow cytometry analysis to examine TCF-1 expression in the WT by CD45.1 or in the TCF-1cKO by CD45.2 bone marrow-derived CD4+ T cells [17]. Our data confirmed that the loss of TCF-1 in the TCF-1 cKO mice is cell-intrinsic, compared to control groups (Appendix A). However, when we examined mixed bone marrow chimera derived CD4 T cells from our chimera model from either WT or TCF-1 cKO mice developed in the same thymus, we did not observe any differences in either CD122 or CD44 expression (Appendix A). Next, we examined whether mixed bone marrow derived CD4 T cells from WT that develop in the same thymus as bone marrow derived CD4 T cells from TCF-1 cKO bone marrow derived model developed in the same thymus, we observed no differences in the EM and CM phenotypes (Appendix A). We also observed no difference in T-bet expression among the mixed bone marrow derived CD4 T cells from WT and TCF-1 cKO mice (Appendix A). The increase in expression of CD122, CD44, EM, CM, and T-bet were significantly higher in naïve CD4 T cells from TCF-1 cKO mice (Figure 1B–E). However, in a mixed chimera model we did not see any differences. These data suggested that these activating marker CD122 and CD44 expression in a mixed bone marrow chimera become similar to that found in TCF-1 cKO mice. EM, CM, and T-bet expression also become similar to TCF-1 cKO mice. Altogether, these data showed that TCF-1 controls a number of activation markers and memory formation in a cell-extrinsic way. 

### 2.2. Loss of TCF-1 in Donor CD4 T Cells Reduces Severity and Persistence of GvHD Symptoms

To investigate whether TCF-1 in mature CD4 T cells contributes to GvHD after allo-HSCT, we employed a murine model of MHC-mismatched allotransplantation. The MHC haplotype mismatch (H2K^b^ in donors, H2K^d^ in recipients) results in the alloactivation of the donor T cells, leading to GvHD [35,54,55]. A group of irradiated BALB/c mice were transplanted with 10 × 10^6^ _TCD_BM cells alone. This group of mice will be considered a control that will not develop GvHD due to the lack of mature T cells. A second group of irradiated BALB/c mice were transplanted with 10 × 10^6^ _TCD_BM along with 1 × 10^6^ CD4 T cells from WT mice. These groups will develop lethal GvHD, as shown previously [35,43,54,55]. To determine whether CD4cre might contribute to the development of CD4 T cell-mediated GvHD, a third group of irradiated BALB/c mice were transplanted with 10 × 10^6^ _TCD_BM along with 1 × 10^6^ CD4 T cells from CD4cre mice. Next, we asked whether CD4 T cells from TCF-1^Flox/Flox^ mice with or without CD4cre develop CD4 T cell- mediated GvHD. A fourth group of irradiated BALB/c mice were transplanted with 10 × 10^6^ _TCD_BM along with 1 × 10^6^ CD4 T cells from TCF-1^Flox/Flox^ mice.

Finally, we examined whether CD4 T cells from mice lacking TCF-1 expression specifically on mature T cells develop GvHD. A fifth group of irradiated BALB/c mice were transplanted with 10 × 10^6^ _TCD_BM along with 1 × 10^6^ CD4 T cells from TCF-1 cKO mice. Recipient BALB/c mice were monitored for survival (Figure 2A) for up to about 70 days. Recipient mice were examined, weighed, and given a GvHD score (Figure 2B,C) to identify the severity of GVHD for up to about 70 days post-transplant. Recipient mice were scored based on weight loss, fur texture, posture, activity, skin condition, and diarrhea, as previously described [17,35,54,55,56,57].

Recipient BALB/c mice receiving donor CD4 T cells from WT C57Bl/6, CD4 cre+/+ or TCF-1^Flox/Flox^ control donor’s cells experienced a rapid increase in GvHD symptoms and peaked at a very high score, indicating very severe disease (Figure 2C). CD4 T cells are known to cause very severe GvHD symptoms, so this finding was expected [35,53,58]. Over time, these recipient mice continued to show severe symptoms, with consistent scores until death from disease (Figure 2C). Survival was also poor, with most mice in this group dying within 25 days of transplantation (Figure 2A). These mice lost weight due to GvHD and died before they were able to regain much weight (Figure 2B). In contrast, recipient BALB/c mice receiving CD4 T cells from TCF-1 cKO C57Bl/6 mice had significantly better survival (Figure 2A), weight gain following an initial weight loss (Figure 2B) and reduced GvHD scores (Figure 2C). Interestingly, recipient BALB/c mice in the TCF-1 cKO CD4 T cell showed a peak in GvHD score early on around day 5, as was seen in the other control group, but peaked at a much lower score. Additionally, this peak score did not persist over time, as the scores for these mice quickly reduced to the level seen in bone marrow-only transplanted controls (Figure 2C). In addition, this low score remained for an extended period, suggesting that the disease had resolved rather than being delayed (Figure 2C). These data indicated that GvHD symptoms were not only less severe in these mice, but also less persistent over time.

### 2.3. TCF-1 Regulates Chemokine/Chemokine Receptor Expression in Mature CD4 T Cells during Allo-Activation

GvHD involves early migration of alloreactive donor T cells into the target organs, followed by T cell expansion and tissue destruction [23,24]. Modulation of alloreactive T cell trafficking has been suggested to play a significant role in ameliorating experimental GvHD [59]. Therefore, we examined the trafficking of donor T cells to GvHD target tissues, as previously described [60]. To determine whether TCF-1 regulates CD4 T cells trafficking to GvHD target organs, we repeated the short-term experiments, as described. Irradiated BALB/c recipient mice were transplanted with 10 × 10^6^ _TCD_BM. We mixed CD4 T cells from WT mice with WT B6LY5 (CD45.1^+^) in a C57Bl/6 background with CD4 T cells from TCF-1 cKO mice with a (CD45.2^+^) C57Bl/6 background at a 1:1 ratio. Seven days post-transplantation, recipient mice were examined for the presence of donor CD4 T cells in the spleen, lymph nodes, liver, and small intestines. We observed no differences in trafficking of donor CD4 T cells from either WT or TCF-1 cKO mice (Appendix A). Chemokines direct cellular infiltration to tissues, and their receptors and signaling pathways represent targets for therapy in multiple disease models, including autoimmunity, cancer, and T cell responses to viral infections [61,62,63,64,65,66]. 

To determine whether TCF-1 regulates specific chemokine receptors expression, we sorted back donor CD4 T cells from allotransplanted BALB/c mice (using H2K^b^ to identify donor cells) and performed a qPCR using a 96-well mouse chemokine/chemokine receptor array plate (Thermo Fisher liver pool NY USA). We found that the expression of chemokines and chemokine receptors were upregulated following alloactivation, as expected. However, expression of these markers was consistently higher in TCF-1 cKO CD4 T cells from spleen, both pre- and post-transplant (Figure 3A,B), while these markers were downregulated in TCF-1 cKO CD4 T cells from post-transplanted liver (Figure 3C). These changes also confirmed our observation of CXCR3 expression as increased in splenic cells from TCF-1-deficient mice, compared to WT mice. Our data uncovered that TCF-1 regulates chemokine receptor expression before and after transplantation, with tissue-specific changes. 

### 2.4. TCF-1 Regulates CD4 T Cell Damage to GvHD Target Organs

During GvHD, host tissues are damaged by the activity of alloactivated CD4 T cells [53,67,68]. To determine whether damage to the target organs of GvHD (skin, liver, and small intestine) was altered by loss of TCF-1 in donor CD4 T cells, we collected organs from mice allotransplanted, as described above [35,54,55,69]. To induce GvHD, we used MHC-mismatched donors and recipients, with T cell-depleted bone marrow (_TCD_BM) from WT mice, donor CD4 T cells from either C57BL/6 (B6) WT or TCF-1 cKO mice (MHC haplotype b), and lethally irradiated BALB/c (MHC haplotype d) mice as recipients. Recipient mice were injected intravenously with 10 × 10^6^ wild-type (WT) _TCD_BM cells along with purified 2 × 10^6^ donor CD4 T cells. To examine the pathological damage to target organs including the liver, small intestine, and skin, tissues from the recipient BALB/c mice and these organs were collected for histology at day 7 and day 21 post-transplantation. Collected organs were fixed, sectioned, hematoxylin and eosin (H&E) stained, and analyzed by a pathologist (L.S.) In the liver, (magnification ×400), much less inflammatory infiltrates in the bile ductal epithelium of the portal triad (black arrows showing the inflammatory cells around interlobular bile ducts) was seen in the recipient mice transplanted with CD4 T cells from TCF-1 cKO cells, compared with recipient mice transplanted with CD4 T cells from WT C57Bl/6 mice, and both euthanized at day 7r: WT (Figure 4A) and TCF-1 cKO (Figure 4B); euthanized at day 21: WT (Figure 4C) and TCF-1 cKO (Figure 4D) post-transplant. In the small intestine (magnification ×400), no apoptotic bodies were seen in the crypts of the small intestine in the recipient mice transplanted with CD4 T cells from TCF-1 cKO C57Bl/6, while many apoptotic bodies with micro abscesses (black arrows and red circle) were present in the small intestine of the recipient mice transplanted with CD4 T cells from WT C57Bl/6 mice and euthanized at day 7: WT (Figure 4E) and TCF-1 cKO (Figure 4F). We observed that fewer apoptotic bodies were present in the small intestine of recipient mice that were transplanted with CD4 T cells from WT C57Bl/6 mice and euthanized at day 21, compared to recipient mice transplanted with CD4 T cells from TCF-1 cKO mice at day 21: WT (Figure 4G) and TCF-1 cKO (Figure 4H). In the skin (magnification ×200), a mild increase of inflammatory cells (red circle) was observed in the dermis of the recipient mice transplanted CD4 T cells from WT C57Bl/6 mice and euthanized on day 7: WT (Figure 4I) and TCF-1 cKO (Figure 4J), and a marked increase of inflammatory cells (red circle) with frequent apoptotic bodies involving both epidermic and dermis was observed in the dermis of the recipient mice transplanted with CD4 T cells from WT C57Bl/6 mice and euthanized at day 21 (Figure 4K), while the dermis of the recipient mice transplanted with CD4 T cells from TCF-1 cKO mice appeared normal at both timepoints (Figure 4L). These findings further support the idea that disease resolves over time and does not persist when recipient mice transplanted with donor CD4 cells from TCF-1 cKO mice. Together, these results indicate that TCF-1 normally contributes to and is indispensable for GvHD damage by T cells, and the loss of TCF-1 reduces its severity and persistence of GvHD.

### 2.5. TCF-1 Regulates CD4 T Cell Survival and Persistence

CD4 T cell survival and persistence functions are critical in both health and disease [70]. The importance of CD4 T cell persistence has been shown in autoimmunity, cancer, viral infection, and several cardiovascular diseases [71,72,73]. Therefore, we sought to examine whether TCF-1 is critical for the survival of peripheral CD4 T cells and for their function. We isolated splenocytes from either control mice, including WT C57Bl/6 or TCF-1 cKO mice and performed an in vitro death and apoptosis assay. These CD4 T cells were either stimulated with 2.5 μg/mL anti-CD3 and 2.5 μg/mL anti-CD28 antibodies for 6, 24, 48, or 72 h in culture or left unstimulated, then were stained for apoptosis and death markers. We did not observe any differences in apoptosis, live cell, or dead cell percentages at 0 h between the strains of mice (Figure 5A). CD4 T cells from TCF-1cKO mice that were stimulated for 6 h or for 24 h had more dead cells (annexin V+, near-IR+) and fewer live cells (annexin V-, near-IR-) than cells from the control WT mice (Figure 5B,C). When stimulated for 48 h, CD4 T cells from TCF-1 cKO mice showed more dead (annexin V+, near-IR+) and apoptotic cells (Annexin+, near-IR-) and fewer live cells (annexin V-, near-IR-), compared to CD4 T cells from WT C57Bl/6 mice (Figure 5D). By 72 h post-stimulation, the frequencies of live, apoptotic, and dead cells from the control mice, WT C57Bl/6, and TCF-1 cKO mice were the same (Figure 5E). These data suggested that TCF-1 is critical for early survival and the persistence function of CD4 T cells.

CD4 T cell exhaustion has been well documented in CD4 T cell responses to viral infections, and the role of TCF-1 in regulating T cell exhaustion during viral infections is also clear [74,75,76]. PD-1 has also been shown to be critical for CD4 T cell function [77]. Thus, we examined whether TCF-1 regulates PD-1 expression on naïve splenic CD4 T cells from WT C57Bl/6 or TCF-1 cKO mice. Splenocytes from TCF-1 cKO or control mice and WT C57Bl/6 mice were isolated and were either stimulated with 2.5 μg/mL anti-CD3 and 2.5 μg/mL anti-CD28 antibodies for 24, 48, or 72 h in culture or were left unstimulated. These cells were stained for PD-1, Ki-67, and TOX. We did not observe any differences in PD-1 expression before or after 24 or 48 h of stimulation (data not shown), but at 72 h post-stimulation, CD4 T cells from TCF-1 cKO mice expressed more PD-1, compared to CD4 T cells from WT C57Bl/6 mice (Figure 5F). This finding suggested that TCF-1 is a critical regulator of PD-1 expression during the late stages of in vitro activation. To examine whether CD4 T cells become exhausted due to the lack of TCF-1 in alloimmunity, as is shown during T cell responses to viral infection, we examined the Ki-67 expression and TOX expression [78,79], in CD4 T cells from WT C57Bl/6 and TCF-1 cKO mice before and after stimulation. Our data uncovered that there were no differences at any timepoint in expression of Ki-67 and TOX among CD4 T cells from WT and TCF-1 cKO mice (Figure 5H,I). 

CD28 receptor provides a critical second signal alongside T cell receptor (TCR) ligation for naïve T cell activation. We and others have shown that TCF-1 is critical for TCR stemness [2,17,80]. Published data have also demonstrated that the lack of CD28 significantly weakens TCR stemness [81]. Thus, we examined whether the CD4 T cells from TCF-1 cKO mice also have reduced CD28 expression. We isolated CD4 T cells from either WT or TCF-1 cKO mice. These freshly isolated CD4 T cells were examined for CD28 expression by flow cytometry. Our data uncovered that CD4 T cells from TCF-1 cKO mice have no CD28 expression, compared to that seen in the WT mice (Figure 5J). These findings highlight that TCF-1 has minimal impact on CD4 T cell exhaustion and proliferation in in vivo studies. These findings suggested that TCF-1 regulates CD28 expression required for CD4 stemness. 

To understand whether TCF-1 might regulate CD4 T cells differently in vitro than in vivo, we transplanted 1 × 10^6^ purified CD4 T cells from either WT or TCF-1 cKO mice into irradiated BALB/c mice to establish an allo-HSCT model, as described above, to assess these changes in vivo. At day 7 post-transplant, recipient BALB/c mice were euthanized and donor H2K^b^+ donor CD4 T cells from the liver and spleen were stained for annexin V and near-IR (apoptotic and dead cell markers). Our data uncovered that there were no differences between the strains in live, apoptotic, or dead CD4 T cells coming from the liver or spleen of the recipients (Figure 6A,B). However, our data demonstrated that there were significantly fewer donor H2k^b^+ CD4 T cells in the spleen and liver of recipients that were transplanted with CD4 T cells from TCF-1 cKO mice, compared to in those given WT cells (Figure 6C,D). We also wanted to determine the expression of Ki-67 and TOX in the in vivo alloactivated CD4 T cells from TCF-1-deficient and WT mice. Again, we did not find any differences in Ki-67 and TOX expression in CD4 T cells from the liver or spleen between the recipients of the two donor strains (Figure 6E–H). These findings highlight that TCF-1 have minimal impact on CD4 T cells exhaustion and proliferation in in vivo studies. These findings suggested that TCF-1 regulates CD28 expression required for CD4 stemness. 

### 2.6. TCF-1 Regulates Serum Levels of Cytokines during Alloimmunity

Production of inflammatory cytokines, eventually culminating in a cytokine storm, is considered a hallmark of CD4 T cell-mediated alloimmunity [25,31,82,83]. Th1 cytokines and cytotoxic mediators are essential for T cells to maintain the GVL effect and kill tumor cells, yet they also lead to the damage of healthy host tissues [84,85,86,87], More specifically INF-γ and TNF-α secretion by donor CD4 T cells are the hall mark of persistence of GvHD mediators [88]. To examine cytokine production by TCF-1 cKO CD4 T cells, we allotransplanted recipient mice, as described above. At day 7 post-transplantation, we also took blood from these recipient mice and obtained serum, which we tested for various cytokines using a LEGEND plex ELISA kit (Biolegend). We uncovered that recipient BALB/c mice transplanted with CD4 T cells from TCF-1 cKO mice had higher serum levels of IFN-γ and TNF-α at day 7 post-transplant than recipient mice transplanted with CD4 T cells from WT mice. However, the level of these proinflammatory cytokines were significantly decreased at day 14 post transplantation (Figure 7A–D). IL-5 and IL-2 have been shown to play critical roles in GvHD [89,90,91]. Thus, we examined whether the levels of IL-2 or IL-5 are impacted by the loss of TCF-1 on mature CD4 T cells. Our data uncovered that recipient BALB/c mice transplanted with CD4 T cells from TCF-1 cKO mice showed increased expression of IL-5 at day 7, compared to recipient mice transplanted with CD4 T cells from WT C57Bl/6 mice. However, at day 14, we did not see significant differences in the serum level IL-5 in recipient mice either transplanted with donor CD4 T cells from WT or TCF-1 cKO mice. Similarly, recipient BALB/c mice transplanted with donor CD4 T cells from TCF-1 cKO mice produced higher serum levels of IL-2 at day 7 post transplantation, however the serum levels dropped at day 14 post transplantation (Figure 7E–H). We observed that recipient mice transplanted with CD4 T cells from TCF-1 cKO mice showed higher serum levels of IL-6 after 7 days post-transplant than mice given WT cells, but the IL-6 expression in these mice later dropped at day 14, such that there was no difference compared to mice given WT cells (Figure 7I,J). These findings provide evidence that recipient BALB/c mice transplanted with CD4 T cells from TCF-1 cKO mice exhibit increased levels of serum cytokines linked to GvHD severity from day 1 to day 14, but these levels reduce to WT levels at day 14 onward. This supports the pattern of resolving the disease severity seen in the in vivo models (Figure 2). Published data have shown that TCF-1 plays a significant role in CD8 T cell-mediated cytokine expression in viral infections [86]. However, the role of TCF-1 in CD4 T cell-mediated Th1 and Th2 cytokine production in alloimmunity has not been defined. Thus, we examined whether TCF-1 regulates CD4 T cell-mediated Th2 cytokines in an allogeneic transplant model. Our data showed that recipient BALB/c mice transplanted with CD4 T cells from TCF-1 cKO mice expressed increased levels of IL-4 and IL-13 at day 7 post transplantation, compared to mice given WT cells. However, levels of both IL-4 and IL-13 in mice given TCF-1 cKO cells decreased at day 14, compared at day 7, while we did not observe any difference in IL-4 and IL-13 levels in between day 7 and day 14 in WT CD4 T cell recipients (Figure 7K–N). This was correlated with the GvHD scores of the same recipient mice as well, suggesting less severe and less persistent GvHD in TCF-1-deficient CD4 T cell transplanted mice. Recipient BALB/c mice transplanted with either TCF-1 cKO or WT cells did not show any differences in IL-10, IL-9, IL-17A, and IL-17F expression (Appendix A). 

These data suggest that allotransplanted TCF-1 cKO CD4 T cells are more activated early in the response but are less active (or less present) later, suggesting a unique mechanism for how TCF-1 modulates cytokine responses during alloimmunity. We also wanted to determine the cellular cytokine production from the donor cells to correlate with the cellular and serum levels of cytokines at day 7 post-transplant. Splenocytes were obtained from recipient mice and were restimulated to induce cytokine production. Cells were restimulated in culture with anti-CD3/anti-CD28 or left unstimulated for 6 h at 37 °C, and Golgiplug was included in the culture media. Then, cells were stained for H2Kb, CD3, CD4, TNF-α, and IFN-γ markers. Our data showed that the production of TNF-α or IFN-γ did not appear to be affected by the loss of TCF-1 (Figure 7P). The difference between cellular cytokine production and serum levels could be due to the lower numbers of CD4+ H2kb+ donor cells present at day 7 post-transplant in the liver and spleen for mice given TCF-1 cKO cells (Figure 6C,D). 

### 2.7. TCF-1 Regulates Key Signaling Pathways in Donor CD4 T Cells

To understand the molecular mechanisms behind the changes we saw in the TCF-1 cKO donor CD4 T cells, and to understand the role of TCF-1 in regulating gene expression in these cells, we employed RNA sequencing. We allotransplanted recipient BALB/c mice with WT or TCF-1 cKO CD3 T cells, as described above. FACS-sorted pre-transplant samples (Pre-Tx) of CD4+ donor cells from WT and TCF-1 cKO mice were taken and stored in TRizol. At day 7 post-transplant, donor T cells were sorted back from the spleen of recipients using H2K^b^, CD3, CD4, and CD8 (Post-Tx samples). 

A principal component analysis (PCA) of pre-transplant and post-transplant samples showed two clusters of samples, WT and TCF-1 cKO, that were clearly separated by principal component 1 (PC1 46% for Pre-Tx, 53% for post-Tx), which suggests that the transcriptomic profile of the TCF-1 cKO CD4 T cells differs from the CD4 T cells from WT mice (Figure 8A,B). Further analysis of pre-transplanted samples identified 812 differentially expressed genes (DEGs, defined by FDR < 0.05 and Log FC = 1) in TCF-1 cKO cells, compared to WT cells, of which 220 were downregulated and 592 were upregulated in CD4 T cells from TCF-1 cKO mice (Figure 8C). We identified 839 DEGs (defined by FDR < 0.05) in post-transplanted CD4 T cells from TCF-1 cKO cells, compared to CD4 T cells from WT mice, of which 356 were downregulated and 483 were upregulated in CD4 T cells from TCF-1 cKO mice (Figure 8D). All DEGs were plotted in the heatmap for pre- and post-transplanted samples, and by using the Spearman correlation method, which is associated with hierarchical clustering, the pre-transplanted and post-transplanted samples were categorized into two clusters (WT and TCF-1 cKO). Gene co-regulation was determined by hierarchical clustering by using the Pearson correlation method with a grouping cutoff (k) of two (Figure 8E,F). Module 2 shows all of the upregulated DEGs and module 1 shows all of the downregulated DEGs in pre-transplanted and post-transplanted samples (Figure 8E,F). A gene ontology (GO) analysis of the pre-transplanted and post-transplanted samples revealed that all of the identified DEGs are involved in a number of biological processes, such as cell death, apoptotic processes, T cell-mediated processes, T cell functions, cytokine production, and response to cytokines, among others. We clustered the pathways that related to the cell death and apoptotic processes in a group and listed them based on the adjusted *p*-value (FDR) for both pre- and post-transplanted samples (Appendix A). Genes that were involved in each pathway are also listed in the tables (Appendix A).

The majority of genes involved in cell death and apoptotic processes were upregulated in the pre- and post-transplanted samples in CD4 T cells from TCF-1 cKO samples, compared to CD4 T cells from WT samples (Figure 8G,H). Interestingly, the β-catenin gene (encoded by Ctnnb1) was significantly upregulated in pre-transplanted samples from donor CD4 T cells from TCF-1 cKO mice, and the WNT-4 gene was significantly upregulated in post-transplanted samples in CD4 T cells from TCF-1 cKO mice, suggesting a compensatory mechanism of upregulation of the β-catenin pathway in the absence of TCF-1 (Figure 8G,H). Even though most of the genes shared were in the cell death and apoptosis-related pathways, some of the genes were unique for each pathway (Appendix A). We also clustered the pathways that related to T cell function and signaling in a group and listed them based on the adjusted *p*-value (FDR) for both pre- and post-transplanted samples (Appendix A). For pre-transplant samples, even though the majority of genes involved in T cell function and signaling were upregulated in CD4 T cells from TCF-1 cKO mice, compared to CD4 T cells from WT mice, interestingly, LAT, LCK, ZAP70, and CD3e genes (which are downstream of the TCR) were downregulated in CD4 T cells from TCF-1 cKO mice, compared to CD4 T cells from WT mice (Figure 8I). When we analyzed the T cell function and signaling-related genes in post-transplanted samples, we observed that most of the genes were downregulated in CD4 T cells from TCF-1 cKO mice than in CD4 T cells from WT mice (Figure 8J), which suggests that alloactivated CD4 T cells from TCF-1 cKO mice having attenuated T cell signaling and T cell responses, compared to T cells from WT mice. Even though most of the genes were shared between the T cell function and signaling-related pathways, some of the genes were unique for each pathway (Appendix A). Pre-transplanted CD4 T cells from TCF-1 cKO mice showed upregulation of the genes involved in cytokine production and cell response to cytokines (Figure 8K, Appendix A) compared to CD4 T cells from WT mice. Post-transplanted CD4 T cells from TCF-1 cKO mice showed downregulation of a number of genes that were involved in cytokine production and responses, such as Ifitm1, JAk3, CD4, CD28, Iκβ, IκG, CD3e, and others, which supported the observed decrease in cytokine production from CD4 T cells from TCF-1 cKO mice (Figure 8L). We also observed that a number of genes involved in chemokine receptor signaling and cell adhesion, such as CCL5, CCL3, CCL4 CXCR5, and CXCR3, are downregulated, and Slit2 is upregulated in CD4 T cells TCF-1 cKO mice, compared to CD4 T cells from WT mice (Figure 8L, Appendix A). It has been shown that Slit2 blocks CXCL12/CXCR4-mediated functional effects in T cells, which is important for HIV infection and viral replication. Altogether, the transcriptomic analysis revealed that TCF-1 regulates the CD4 T cell genetic profile, with a loss of TCF-1 directing the cell towards decreased T cell signaling, decreased cytokine and chemokine signaling, and increased apoptosis and cell death, specifically after allotransplantation. 

## 3. Discussion

T cell factor-1 (TCF-1) is a T cell transcription factor that is known to be critical for T cell development, activation, and in some cases, responses to pathogens [1,92,93]. The functional and development role of TCF-1 has been extensively studied in CD8 T cell responses to viral infections [2,17,18,76,94,95]. To some extent, the role of TCF-1 has been examined in CD4 T cell development [5,8,11], however, it is unclear whether TCF-1 may regulate alloactivated CD4 T cells during responses to alloantigens. The main significance of our results is that we utilized a clinically relevant model of allo-HSCT, enabling us to study all of the major CD4 T cell functions, as well as phenotypes, clinical outcomes, and gene expression, in a single model. Several publications, including our own, have shown that CD4 T cells with higher CD44, CD122, and Eomes, or T-bet referred to as the innate memory phenotype (IMP) [54,96] T cells with the IMP significantly delayed the development of GvHD, but were able to clear tumors [35,54,55,97]. Our data also showed that CD4 T cells from TCF-1 cKO mice have a significantly higher expression of a IMP phenotype, suggesting the TCF-1 might regulate the IMP phenotype and TCF-1 is considered a repressor factor for IMP cells. Memory phenotypes have been reported to play a significant role in the induction (or lack thereof) of GvHD [44]. Our data showed that CD4 T cells from TCF-1 cKO mice upregulate the effector or central memory phenotypes, and these mice show a decreased naïve cell population, which suggests that TCF-1 regulates the memory formation of CD4 T cells. We and others have shown that the upregulation of EM and CM plays a central role in GvHD development. Studies have shown that T cells with a higher EM and CM do not cause GvHD. [35,35,44,54]. Thus, our studies are of great importance because they show that TCF-1 regulates both EM and CM on CD4 T cells. 

Another molecule that is critically important for effector memory cell survival and maintenance is ICOS [44,45,98]. The lower expression in ICOS on effector memory CD4 T cells from TCF-1 cKO mice suggest that even though TCF-1-deficient CD4 T cells have more effector memory cells, their survival and homeostasis may be affected by the loss of TCF-1. 

Modulation of alloreactive CD4 T cell trafficking has been suggested to play a significant role in ameliorating experimental GvHD [99]. However, we did not observe any differences in the donor CD4 T cell migration to GvHD target organs, including the liver and small intestine. Pro-inflammatory conditioning treatment may promote T cell migration into GvHD target tissues [100,101]. Donor CD4 T cells upregulate the chemokine receptor expression upon alloactivation, which mediates donor T cells migration to the site of inflammation [102]. Since chemokine receptor expression in T cells is central to several T cell-mediated diseases [43,103], determining whether TCF-1 regulates chemokine expression either positively or negatively is critically important. We uncovered that mature splenic CD4 T cells from TCF-1 cKO mice expressed higher levels of chemokine receptors than CD4 T cells from WT mice, both pre- and post-allo-HSCT. However, our data showed that CD4 T cells from the liver in mice given TCF-1 cKO cells showed reduced chemokine receptor expression post-transplantation. Our data showed that CD4 T cells from TCF-1 cKO mice caused significantly less tissue damage. All of our data suggest that CD4 T cells from TCF-1 cKO mice can migrate to GvHD target organs, but also provided stronger evidence that TCF-1 is critical for CD4 T cell stemness, however due to the loss of TCF-1 on CD4 T cells, these CD4 T cells are unable to cause persistence of GvHD symptoms. TCF-1 has been shown to be significantly important in T cell development and survival [104], so we also examined whether TCF-1 is critical for CD4 T cell survival in both in vitro and in vivo models. Our data showed that CD4 T cells from TCF-1 cKO mice developed more rapid cell death and apoptosis in vitro within the first 48 h. We observed increased PD-1 expression on TCF-1 cKO cells versus WT cells only at 72 h of in vitro culture, indicating that the cells that survived after 72 h might cause less severe CD4 T cell-mediated diseases [77,105]. Even though we did not observe any differences in dead, apoptotic, or live cell percentages in in vivo alloactivated CD4 T cells from TCF-1 cKO or WT mice, the frequency of donor CD4+ H2kb+ T cells from TCF-1 cKO mice was significantly less in GvHD target organs, thus supporting our central hypothesis that TCF-1 is indispensable for CD4 T cell stemness. However, our recent findings suggested that TCF-1 is dispensable for anti-tumor response [17]. TCF-1 has been shown to play a critical role in CD4 T cell exhaustion and activation in responding to viral infections [75,76]. Our data, both in vivo and in vitro, showed that activated CD4 T cells from TCF-1 cKO mice had no change in exhaustion. These findings demonstrate that alloactivated CD4 T cells are functioning significantly different to how CD4 T cells from TCF-1 cKO mice function in response to viral infections. 

Pro-inflammatory cytokine production by donor cells and host tissues causes damage to nearby healthy host cells [25,84,87]. Our data showed that during initial activation, donor CD4 T cells from TCF-1 cKO mice produce more serum level cytokines, but these drop significantly over time. These findings suggest that despite early increased activation, cytokine production by donor CD4 T cells from TCF-1 cKO mice donor cells quickly reduce post-transplant, allowing the disease to resolve, further confirming that TCF-1 is required for CD4 T cell persistence functions, including cytokine production. This supports our hypothesis that donor CD4 T cells from TCF-1 cKO mice become exhausted and stop proliferating and producing cytokines, allowing for the resolution of the usual persistent disease state. Even though, we initially (day7) observed significant increases in proinflammatory cytokines, including TNF-a and IFN-g from CD4 T cells from TCF-1 cKO mice, this upregulation of TNF-a and IFN-g fails to recruit other inflammatory cells, such as macrophages to the site of inflammation to induce GvHD. These findings suggest that the loss of TCF-1 significantly weakens CD4 T cell persistence during GvHD by the loss of CD28 on CD4 T cells. These findings are supported by our recent publication that the loss of TCF-1 significantly weakens TCR signaling on CD8 T cells [17].

Our transcriptomic data uncovered that TCF-1 regulates several pathways that are critical for CD4 T cell-mediated diseases. Cell death pathways play a central role in CD4 T cell-mediated diseases, including autoimmunity and cancer [17]. We uncovered that TCF-1 significantly and differentially regulates cell death and apoptotic process-related pathways before and after transplantation, which is critical to understand the role of TCF-1 in alloimmunity and fighting infection and cancer. Our data showed that TCF-1 significantly regulates genes in CD4 T cell programmed cell death, including pathways for apoptotic signaling, necrotic signaling and mitochondrial fragmentation. CD4 T cell activation, signaling proliferation, and Th1/Th2/Th17 differentiation are central to both CD4 T cell function and CD4 T cell-mediated diseases, including CD4 T cell responses to viral infection, autoimmunity, cancer, and aging [106].

We also observed that TCF-1 differentially regulates sets of genes in the I-κβ and NF-κβ pathways. This information will enable us to develop target specific approaches to design therapeutic interventions. CD4 T cells primarily function as regulators of other immune cells either through secreted cytokines or by direct cell–cell contact. Inflammatory and anti-inflammatory cytokines production are central to T cell responses to viral infections, autoimmune disorders, cancer, and GvHD [30]. 

Our data uncovered that CD4 T cells from TCF-1 cKO mice have significantly higher β-catenin expression before transplantation and higher WNT4 expression after transplantation, compared to cells from WT mice. Another key finding of this report is that TCF-1 regulates CD28 expression. CD28 is a key co-stimulatory receptor that plays a central role in T cell receptor stemness [2,80]. Published data has also demonstrated that the lack of CD28 significantly weakens TCR stemness [81]. Therefore, both our in vivo and in vitro data demonstrated that CD4 T cells from TCF-1 cKO mice are prone to activation and cell death. These findings are consisting with transcriptomic data and the development of apoptosis. These findings are also supported our GvHD studies that CD4 T cells from TCF-1 cKO mice showed peak GvHD clinical scores, but that this significantly diminished over time.

Overall, our data uncovered several novel discoveries regarding how TCF-1 differentially regulates CD4 T cell functions, at baseline and during alloactivation. More significantly, how TCF-1 functions during T cell development and on mature peripheral CD4 T cells was not previously known for an alloactivation context. These discoveries will enable us to design target specific approaches in treating CD4 T cell-mediated diseases and alloimmunity. 

Limitations of the Study: Currently, the limitation of this study is the use of a mouse model. We are working with structural and medicinal chemists to make specific activators for Wnt/β-catenin pathways. Currently available reagents are Wnt3 ligands [107] or GSK3β-inhibitors. The primary problems with these activators are that either T cells become over activated or there is non-specific activation of several other signaling proteins. Therefore, we are currently working to develop our own specific activators.

## 4. Materials and Methods

### 4.1. Mice

Thy1.1 (B6.PL-Thy1a/CyJ, 000406), B6-Ly5 (CD45.1+, AKA “WT” or B6.SJL-Ptprc^a^ Pepc^b^/BoyJ, 002014), and BALB/c mice (CR:028) were purchased from Charles River or Jackson Laboratory. TCF-1 cKO mice (Tcf7 flox/flox cross bred with CD4cre) [108] were obtained from Dr. Jyoti Misra Sen at the NIH and bred in our facilities. CD4cre (022071). Using genomic PCR, we confirmed that our newly generated mice are TCF-1 cKO. Eight–12-week-old and sex-matched mice were used for all experiments. Recipient mice for transplant experiments were female BALB/c mice (CR:028 from Charles River, age 8 weeks or older). Recipient mice for the chimera experiments were Thy1.1 mice (B6.PL-Thy1a/CyJ, 000406 from Charles River). Animal maintenance and experimentation were approved by the Upstate Medical University IACUC committee with IACUC #433. All mice used for transplants were female, and flow cytometry experiments were carried out with both male and female mice. 

### 4.2. DNA Extraction and PCR

Donor mice were genotyped using PCR. Ear punches were taken from each mouse at 4 weeks of age, DNA was extracted, and run in a PCR reaction using the Accustart II mouse genotyping kit (95135-500 from Quanta Biosciences). Standard PCR reaction conditions and primer sequences from Jackson Laboratories were used for Eomes, T-bet, and CD4 cre+/+. For TCF-1, primer sequences and reaction conditions were obtained from Dr. Jyoti Misra Sen of NIH.

Primers used for CD4 cre+/+ genotyping are: Common primer: 5′-GTTCTTTGTATATATTGAATGTTAGCC; WT reverse primer: 5′-TATGCTAGGACAAGAATTGACA; and Mutant reverse primer: 5′-CTTTGCAGAGGGCTAACAGC. PCR conditions: Step 1. 94 °C for 2:00 min; Step 2. 94 °C, 20 s; Step 3. 65 °C, 15 s; Step 4. 68 °C, 10 s; Step 5. Go to Step 2, 10×; Step 6. 94 °C, 15 s; Step 7. 60 °C, 15 s; Step 8. 72 °C, 10 s; Step 9. Go to Step 6, repeat 28×; Step 10. 72 °C, 2 min; Step 11. 10 °C, infinite hold.

Primers used for TCF-7 genotyping: Forward primer: 5′-AGCTGAGCCCCTGTTGTAGA, Reverse primer #1: 5′-TTCTTTGACCCCTGACTTGG, Reverse primer #2: 5′-CAACGA GCTGGGTAGAGGAG. PCR conditions for TCF-7 are: Step 1. 94 °C, 2 min; Step 2. 55 °C, 30 s; Step 3. 72 °C, 1 min; Step 4. Go to Step 2. repeat 38×; Step 5. 72 °C, 10 min, 12 °C infinite hold.

### 4.3. Flow Cytometry, Sorting, and Phenotyping

For phenotyping experiments, splenocytes were obtained from WT C57Bl/6 and CD4 cre+/+ C57Bl/6 control mice and TCF-1 cKO mice. For all other experiments, cells were obtained from transplanted recipients. Cells were incubated with RBC lysis buffer (00-4333-57 from eBioscience) to remove red blood cells when necessary. Following processing, cells were stained in MACS buffer (1× PBS with EDTA and 4 g/L BSA) with extracellular markers and were incubated for 30 min on ice. Cells were then washed and run on a BD LSRFortessa flow cytometer to collect data. If intracellular markers were used, cells were washed after extracellular staining, then fixed overnight using buffers from the Fix/Perm Concentrate and Fixation Diluent from FOXP3 transcription factor staining buffer set (eBioscience cat. No. 00-5523-00). The following day, cells were washed in Perm buffer from the same kit and were stained with intracellular markers in Perm buffer for 40 min at room temperature. Stained cells were resuspended in FACS buffer (eBioscience cat. No. 00-4222-26) and transferred to flow tubes. All antibodies were used at 1:100 dilution and were purchased from Biolegend or eBioscience. The cells were then washed and run on a BD LSRFortessa. For cell sorting, cells were stained in the same manner and run on a BD FACSAria, equipped with cold-sorting blocks. Cells were sorted into sorting media (50% FBS in RPMI) for maximum viability, or TRizol for RNAseq/qPCR experiments. All flow cytometry data was analyzed using FlowJo software v9 (BD). Depending on the experiment, the antibodies used were: anti-CD4 (FITC, PE, BV785, BV510, BV421), APC, PerCP, Pacific Blue, PE/Cy7), anti-CD3 (BV605 or APC/Cy7), anti-H2Kb-Pacific Blue, anti-H2Kd-PE/Cy7, anti-CD122 (FITC or APC), anti-CD44 (APC, PercP or Pacific Blue), anti-CD62L-APC/Cy7, anti-ICOS-PE, anti-CXCR3-Percp-Cy5.5, anti-TNF-α-FITC, anti-IFN-γ-APC, anti-Eomes (AF488 or PE/Cy7), anti-T-bet-BV421, anti-CD45.2-PE/Dazzle594, anti-CD45.1-APC, anti-Ki67 (PE or BV421), anti-PD-1-BV785, anti-annexin V-FITC, LIVE/DEAD near IR, and anti-TOX-APC. 

### 4.4. Bone Marrow Transplants

#### 4.4.1. Allo-HSCT and GVHD Studies

For GvHD experiments, we utilized the MHC-mismatch mouse model of allo-HSCT (WT C57Bl/6 ➔ BALB/c, i.e., H2K^b+^ ➔ H2K^d+^). BALB/c recipient mice were irradiated twice with 400 cGy x-rays (total dose 800 cGy), with a rest period of at least 12 h between doses, and 4 h of rest prior to transplantation. Lethally irradiated BALB/c mice were transplanted with 10 × 10^6^ T cell-depleted bone marrow (_TCD_BM) cells. Briefly, bone marrow T-cells were depleted using CD90.2 MACS beads (130-121-278 from Miltenyi) and LD columns (130-042-901 from Miltenyi) [35,54]. T cells CD4+ were separated from donor mice spleens using CD90.2 or CD4 microbeads and LS columns (Miltenyi, CD4: 130-117-043, CD90.2: 130-121-278, LS: 130-042-401) [17,35]. We used purified donor CD4 T cells from WT, CD4cre, or Tcf7 flox/flox mice as controls and TCF-1 cKO mice as the experimental group. The cells were then IV injected into the tail vein in PBS. The recipient mice received 1 × 10^6^ T cells per mouse, along with 10 × 10^6^ T cell-depleted bone marrow cells (_TCD_BM) collected from WT C57BL/6J mice. Recipient mice were evaluated for clinical signs of GvHD and weight loss for more than 70 days [17,35]. Clinical presentation of the mice for each experiment was assessed 2–3 times per week by a scoring system that summed the changes in 6 clinical parameters: diarrhea, posture, activity, fur texture, weight loss, and skin integrity (Cooke et al., 1996). Mice were euthanized when they lost ≥ 30% of their initial body weight [17,35,56]. 

#### 4.4.2. Allo-HSCT Sort Term Experiments

Lethally irradiated BALB/c mice were transplanted, as described above. At day 7 or day 14, the recipients were euthanized, and serum, spleen, small intestine, skin, or liver was collected, depending on the experiment. 

#### 4.4.3. Bone Marrow Chimera Model

For the mixed bone marrow chimera experiments, a 1:4 ratio of WT (CD45.1) to TCF-1 cKO (CD45.2) bone marrow was injected into Thy1.1 mice, and the mice were rested for 9 weeks. This ratio was chosen based on our previous publication [54]. At 9 weeks post-transplant, tail vein blood was collected and stained with anti-CD45.1 and anti-CD45.2 to detect the two donor cell types. At 10 weeks, these chimeras were euthanized, and their spleens were processed and stained for phenotyping by flow cytometry.

### 4.5. qPCR Analysis

Mice were short-term transplanted, as described above (1 × 10^6^ donor CD4 T cells and 10 × 10^6^ _TCD_BM) cells, and at day 7, recipient mice were humanely euthanized. Splenocytes were obtained from pre- and post-transplanted mice, and FACS sorted, as described above to obtain donor cells. These cells were all sorted into TRizol, then RNA was extracted using chloroform phase separation protocols. The extracted RNA was eluted using the Qiagen RNAEasy Minikit (74104 from Qiagen Germantown MD USA) and run on a spectrophotometer to determine concentration. RNA was then converted to cDNA with the Invitrogen Super-Script IV First Strand Synthesis System kit (18091050 from Invitrogen) and run on a spectrophotometer to determine the concentration. The master cocktail, including 10 ng/μL cDNA and Taqman Fast Advanced Master Mix (4444557 from Invitrogen), was prepared for each sample, and 20μL was added to each well of a 96 well custom TaqMan Array plate with chemokine/chemokine receptor primers (Thermo Fisher, Mouse Chemokines & Receptors Array plate, 4391524). The plates were run on a Quantstudio 3 thermocycler, according to manufacturer’s instructions for the TaqMan assay, and data were analyzed using the Design and Analysis software v2.4 (provided by Thermo Fisher). Five separate recipient mice were sorted, and cells were combined to make one sample for qPCR testing per condition/organ. 

### 4.6. Histopathological Evaluation

Lethally irradiated recipient mice were transplanted with 10 × 10^6^ _TCD_BM) cells, 1 × 10^6^ WT CD4^+^ T donor cells and at day 7 and day 21 post-transplant, the organs were removed from WT or TCF-1 cKO T cell-transplanted recipient mice. The spleen, liver, small intestine, and skin from the back and ear were removed and fixed in 10% neutral buffered formalin. Tissue pieces were sectioned and stained with hematoxylin and eosin (H&E) by the Histology Core at Cornell University. Stained slides were then imaged and analyzed by a pathologist at SUNY Upstate (L.S.) who was blinded to the study conditions and slide identity.

### 4.7. Isolation of Lymphocytes from the Liver and Small Intestine

To isolate the lymphocytes from the liver, they were perfused with 5–10 mL of ice-cold PBS to remove red blood cells (RBCs) before removal. Livers were then ground through a 70 mm filter with PBS, centrifuged to remove debris, and lymphocytes were isolated by a 22-min spin in 40% Percoll in RPMI/PBS (22 °C, 2200 rpm, no brake, no acceleration). Isolated lymphocyte pellets were washed, cells were briefly incubated with red lysis buffer to remove remaining RBCs, and resuspended with PBS or MACS buffer (BSA in PBS). To isolate lymphocytes from the small intestine, the intestine was removed and put into an ice cold MACS buffer, opened lengthwise, washed with MACS, and epithelial cells were stripped off by a 30 min shaking incubation (37 °C) in strip buffer (1× PBS, FBS, EDTA 0.5 M, and DTT 1 M). The guts were then cut into small pieces and digested by a 30 min shaking incubation (37 °C) in digestion buffer (collagenase, DNAse, and RPMI). The tubes were then vortexed, and liquid and solid gut pieces were filtered through a 70 mm filter to obtain a cell suspension. Percoll was then used to isolate lymphocytes, as was carried out for liver, with no RBC lysis afterwards. The gut cells were then placed in MACS buffer for further use. 

### 4.8. Cellular Level Cytokine Production Assay

Mice were short-term transplanted, as described above (1 × 10^6^ donor CD4 T cells), and at day 7 post-transplant, recipient mice were humanely euthanized and splenocytes were obtained. The splenocytes were cultured for 6 h at 37 °C and 7% CO_2_ with GolgiPlug (1:1000) and PBS or anti-CD3 (1 μg/mL)/anti-CD28 (2 μg/mL) to restimulate them. Then, after 6 h, the cells were removed from the culture, stained for surface markers, fixed and permeabilized, then stained for the cytokines IFN-g and TNF-a using the BD Cytokine Staining kit (555028), and run on a flow cytometer.

### 4.9. Serum Level Cytokine Production Assay

Mice were short-term transplanted, as described above with 10 × 10^6^ _TCD_BM) cells, along with 1 × 10^6^ WT CD4^+^ T donor T cells. At day 7, the recipient mice were euthanized, and serum was obtained from cardiac blood. The serum was collected from recipient mice in the cytokine experiment and analyzed using the Biolegend LEGENDplex Assay Mouse Th Cytokine Panel kit (741043) [35,54]. This kit quantifies serum concentrations of: IL-2 (T cell proliferation), IFN-g and TNF-a (Th1 cells, inflammatory), IL-4, IL-5, and IL-13 (Th2 cells), IL-10 (Treg cells, suppressive), IL-17A/F (Th17 cells), IL-21 (Tfh cells), IL-22 (Th22 cells), IL-6 (acute/chronic inflammation/T cell survival factor), and IL-9 (Th2, Th17, iTreg, Th9—skin/allergic/intestinal inflammation). Data were collected on a BD LSR Fortessa cytometer, and data were analyzed using the LEGENDplex software (provided with kit via Biolegend).

### 4.10. Cell Death Assay

#### 4.10.1. In Vitro

We obtained splenocytes from WT and TCF-1 cKO mice and either activated them with 2.5 μg/mL anti-CD3 (Biolegend #100202) and anti-CD28 antibodies (Biolegend #102115) for 6, 24, 48, or 72 h in culture, or left them unstimulated. Annexin V-FITC (V13242 from Invitrogen) and LIVE/DEAD near IR (L34976 from Invitrogen) were used to identify dead (Ann.V+NIR+), live (Ann.V-NIR-), and apoptotic (Ann.V+NIR-) T cells 

#### 4.10.2. In Vivo

Mice were short-term transplanted, as described above (1 × 10^6^ CD4 donor T cells), and at day 7 post-transplant, cells from the spleen and liver were stained with annexin V-FITC (V13242 from Invitrogen) and LIVE/DEAD near IR (L34976 from Invitrogen). Annexin V and NIR were used to identify dead (Ann.V+NIR+), live (Ann.V-NIR-), and apoptotic (Ann.V+NIR-) cells. Donor T cells were identified by H2Kb, CD3, and donor CD4 T cells. 

### 4.11. RNA Sequencing

Freshly isolated CD4T cells were FACS sorted in TRizol for pre-transplanted (Pre-Tx) samples. For post-transplanted (post-Tx) samples, recipient mice were transplanted with 1 × 10^6^ CD3 donor T cells, and at day 7 post-transplant, donor CD8 T cells were FACS-sorted back from the recipient spleen of TCF-1 cKO and WT transplanted mice and sorted into TRizol. RNA was extracted from all of the pre- and post-transplanted samples and prepped by the Molecular Analysis Core (SUNY Upstate, https://www.upstate.edu/research/facilities/molecular-analysis.php (accessed on 21 November 2022)). Paired end sequencing was carried out with an Illumina NovaSeq 6000 system at the University at Buffalo Genomics Core (http://ubnextgencore.buffalo.edu (accessed on 21 November 2022)). The statistical computing environment R (v4.0.4), the Bioconductor suite of packages for R, and Rstudio (v1.4.1106) were used for transcriptomic analysis. Kallisto (version 0.46.2) was used for transcript abundance determination and performing the pseudoalignment. Calculated transcript per million (TPM) values were normalized and fitted to a linear model by the empirical Bayes method with the Voom and Limma R packages to identify differentially expressed genes (DEGs) for both pre- and post-transplanted samples. For pre-transplant samples, DEGs filtered by adjusted *p*-value (FDR) < 0.05, log fold change = 1, and for post-transplant samples by adjusted *p*-value (FDR) < 0.05. DEG’s were used for hierarchical clustering and heatmap generation in R. A gene ontology enrichment analysis was conducted using the g: Profiler toolset; g:GOSt tool. Data will be deposited in the Gene Expression Omnibus (GEO) database for public access (https://www.ncbi.nlm.nih.gov/geo (accessed on 21 November 2022)). With accession number GSE204747.

### 4.12. Statistical Analysis

All numerical data are reported as means with standard deviation unless otherwise noted. Data were analyzed for significance with GraphPad Prism v9. Differences were determined using one-way ANOVA and Tukey’s multiple comparisons tests for three or more groups, or with a Student’s *t*-test when only two groups were used. A Kaplan–Meier survival analysis was used for survival experiments. All tests were two-sided. *p*-values less than or equal to 0.05 were considered significant. All transplant experiments were carried out with N = 3–5 mice per group and repeated at least twice. Ex vivo experiments were carried out two to three times unless otherwise noted with at least three replicates per condition each time. RNAseq was carried out once with three replicates per group. qPCR was completed once with one sample per condition, and 5 mice combined to make the one sample. Sorting was carried out once for each of these two experiments, and data were recorded for Figure 3 and Figure 8. Data in the figures are presented as mean and SD unless otherwise noted. 

## Figures and Tables

**Figure 1 ijms-24-04326-f001:**
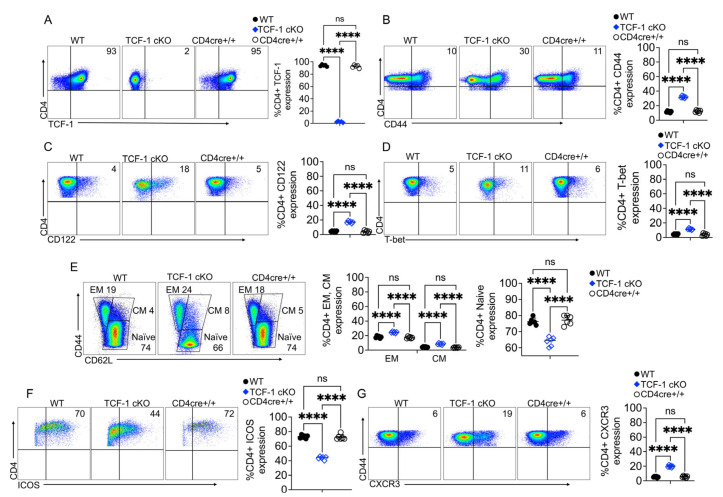
TCF-1 regulates CD4 T cell phenotypes and memory formation. (**A**–**G**) Splenocytes from WT, CD4cre+/+ or TCF-1 cKO donor mice were isolated, and stained for flow cytometry, and run on a BD LSRFortessa flow cytometer. (**A**) Flow plot of CD4 T cells expressing TCF-1 and statistical analysis of the percent of CD4 T cells expressing TCF-1 with a quantitative analysis. (**B**) Flow plot of CD4 T cells expressing CD44 and a statistical analysis of the percent of CD4 T cells expressing CD44 with a quantitative analysis graph. (**C**) Flow plot of CD4 T cells expressing CD122 and a statistical analysis of the percent of CD4 T cells expressing CD122 with a quantitative analysis graph. (**D**) Flow plot of CD4 T cells expressing T-bet and a statistical analysis of the percent of CD4 T cells expressing T-bet. Quantitative analysis of several mice presented as a graph. (**E**) Flow plot of the memory phenotype of CD4 T cells by expression of CD44 and CD62L and a statistical analysis of the percent of CD4 T cells with memory phenotypes. Quantitative analysis of several mice presented as a graph. (**F**) Flow plot of effector memory CD4 T cells expressing ICOS and a statistical analysis of the percent of the effector memory CD4 T cells expressing ICOS. (**G**) Flow plot of CD4 T cells expressing CXCR3 and a statistical analysis of the percent of CD4 T cells expressing CXCR3. All data are shown as individual points with mean and SD, all data were analyzed with Student’s *t*-test or one-way ANOVA (depending on data groups) followed by Tukey’s multiple comparisons. For naïve mice n = 3–5 per group of mice are shown. For chimeric mice n = 5 per group of mice and one experiment is shown (carried out once). **** means *p*-value ≤ 0.0001.

**Figure 2 ijms-24-04326-f002:**
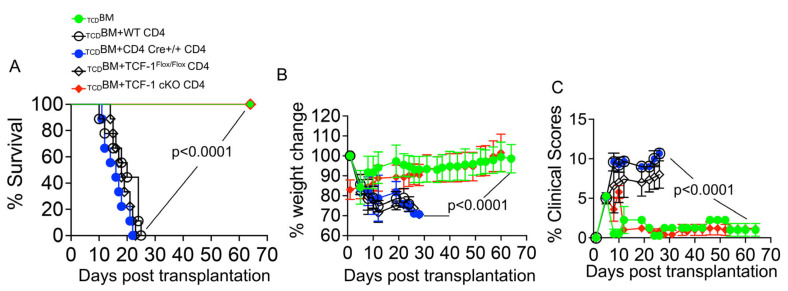
Loss of TCF-1 in donor CD4 T cells reduces severity and persistence of GvHD symptoms. Recipient BALB/c mice (MHC haplotype d) were lethally irradiated and allotransplanted with 10 × 10^6^ WT C57Bl/6 _TCD_BM cells and 1 × 10^6^ CD4 T cells from WT, CD4cre+/+, TCF-1^Flox/Flox^ or TCF-1 cKO donor mice (C57Bl6 background, MHC haplotype b). The mice were weighed and given a GvHD clinical score three times per week for almost 70 days. Score was determined by the combined scores for fur texture, activity level, posture, skin integrity, weight loss, and diarrhea. (**A**) Survival of mice in each group over the 70 day experiment, analyzed with Kaplan–Meier survival statistics. (**B**) Changes in weight, as determined three times a week over 70 days. (**C**) Clinical scores were determined three times a week over 70 days. Mean and SD plotted, analyzed by one-way ANOVA. n = 4 mice/group for BM alone: n = 5 experimental mice/group for all other groups. Survival is a combination of two experiments. Each experiment is repeated twice.

**Figure 3 ijms-24-04326-f003:**
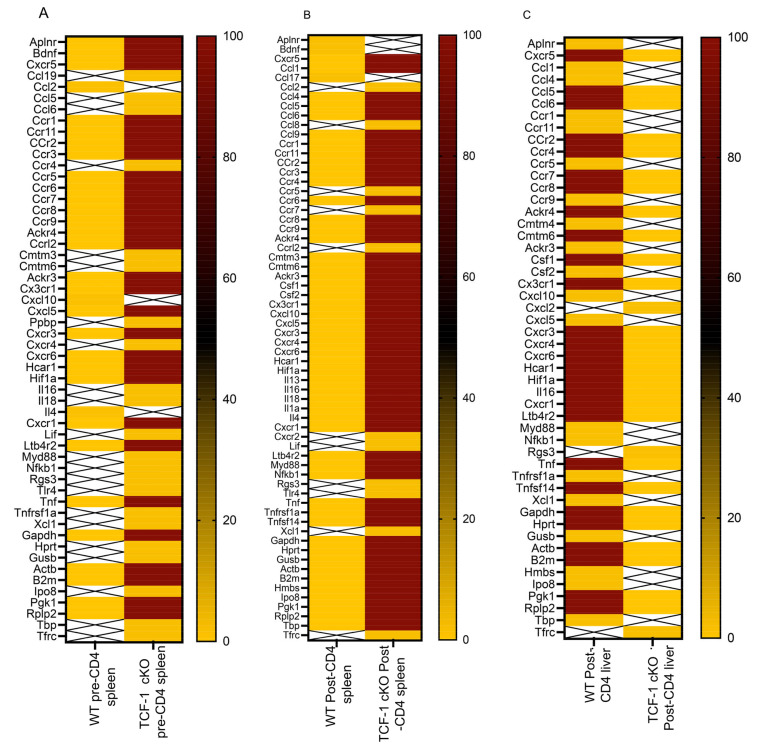
TCF-1 regulates chemokine/chemokine receptor expression in mature CD4 T cells during alloactivation. To perform the qPCR, BALB/c mice were lethally irradiated and allotransplanted, as described above. Donor CD4 T cells from WT and TCF-1 cKO mice were FACS sorted both pre- and post-transplant (7 days after transplant), from the spleen (both pre and day 7) and liver (day 7 only). The cells were sorted in TRizol, and RNA was extracted using chloroform and converted into cDNA using a synthesis kit (more details in Methods). cDNA was added to premade mouse chemokine/chemokine receptor primer plates (Invitrogen Carlsbad CA USA) and run on a Quant Studio 3 thermocycler. Results are shown as a heatmap of fold change per gene, compared to 18S reference gene for each plate. Boxes with an “X” represent signals too low to detect or otherwise unreadable due to technical error. Heatmaps compare WT versus TCF-1 cKO CD4 T cells in (**A**) pre-transplant spleen, (**B**) post-transplant spleen and (**C**) post-transplant liver. One sample was used per donor type and condition, with n = 5 mice into one sample per condition for (**B**,**C**), carried out once. Each experiment was performed twice, one experiment of two experiments is presented.

**Figure 4 ijms-24-04326-f004:**
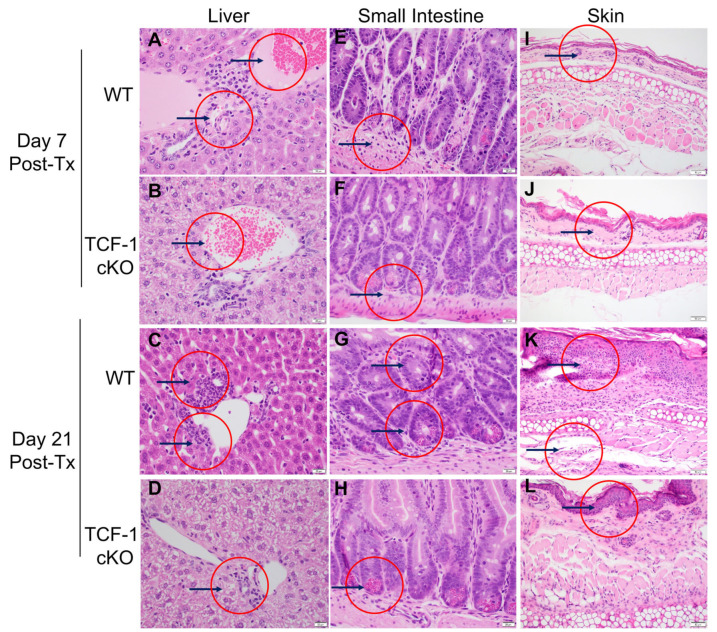
TCF-1 regulates CD4 T cells damage to GVHD target organs. We collected organs from mice allotransplanted, as described above. At day 7 and day 21 post-transplant, organs were taken from recipient mice for histology analyses. Skin, liver, and small intestine (SI) were sectioned, stained with H&E, and analyzed by a pathologist. Representative sections for each organ per group and timepoint are shown. Day 7—H&E liver image of recipient BALB/c mice transplanted with CD4 T cells from (**A**) WT mice and (**B**) TCF-1 cKO mice. Day 21—Liver of recipient mice transplanted with CD4 T cells from (**C**) WT mice and (**D**) from TCF-1 cKO mice. Day 7—SI of the recipient mice transplanted with CD4 T cells from (**E**) WT mice and (**F**) from TCF-1 cKO mice. Day 21—SI of the recipient mice transplanted with CD4 T cells from (**G**) WT mice and (**H**) from TCF-1 cKO mice. Day 7—Skin of the recipient mice transplanted with CD4 T cells from (**I**) WT mice and (**J**) from TCF-1 cKO mice. Day 21- Skin of the recipient mice transplanted with CD4 T cells from (**K**) WT mice and (**L**) from TCF-1 cKO mice. Arrows show lymphocyte infiltration and tissue damage One of the two experiments is presented.

**Figure 5 ijms-24-04326-f005:**
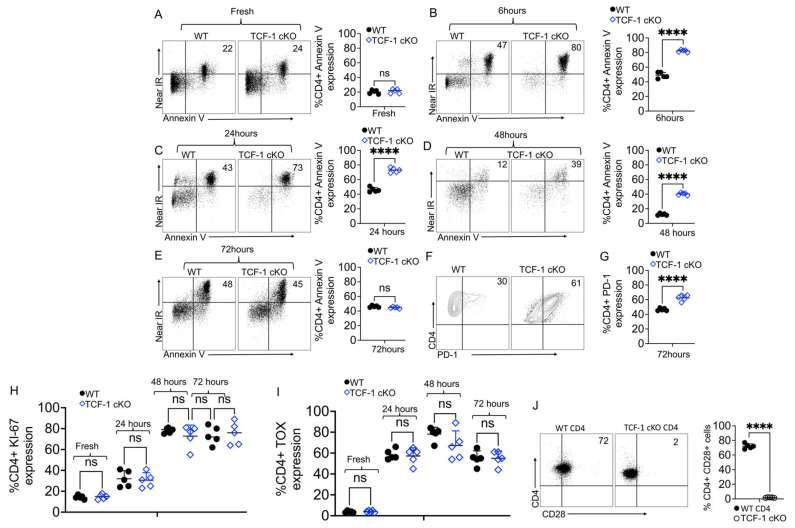
TCF-1 regulates CD4 T cell survival and persistence. (**A**–**E**) Splenocytes from either WT or TCF-1 cKO mice were isolated and either activated with anti-CD3/CD28 antibodies for 6, 24, 48, or 72 h in culture or left unstimulated, and cells were stained for Annexin V and Near IR to determine the dead, apoptotic and live cells. (**A**) Flow plots of CD4 T cells from either WT or TCF-1 cKO mice with quantitative and statistical analyses are shown at 0 h of the dead, apoptotic, and live cells in unstimulated (**B**) CD4 T cells from either WT or TCF-1 cKO mice were stimulated with CD3/CD28, at 6 h with quantitative and statistically analyzed. (**C**) CD4 T cells from WT or TCF-1 cKO mice stimulated with CD3/CD28 for 24 h, shown with quantitative analyses (**D**), CD4 T cells stimulated with CD3/CD28 for 48 h, shown with statistical analyses (**E**) CD4 T cells stimulated with CD3/CD28 for 72 h, shown with statistical analyses. (**F**,**G**) Splenocytes from TCF-1 cKO mice and WT mice were isolated and either activated with anti-CD3/CD28 antibodies for 24, 48, or 72 h in culture or left unstimulated, and were stained for PD-1. (**H**) Splenocytes from TCF-1 cKO mice and WT mice were isolated and either activated with anti-CD3/CD28 antibodies for 24, 48, or 72 h in culture or left unstimulated, and were stained for Ki-67 with quantitative analyses are presented. (**I**) Splenocytes from TCF-1 cKO mice and WT mice were isolated and either activated with anti-CD3/CD28 antibodies for 24, 48, or 72 h in culture or left unstimulated, and were stained for TOX with quantitative analyses are presented. (**J**) CD8 T cells from WT, or TCF-1 cKO mice were examined for CD28 expression. n = 4, one representative of 3 experiments is shown. **** means *p*-value ≤ 0.0001.

**Figure 6 ijms-24-04326-f006:**
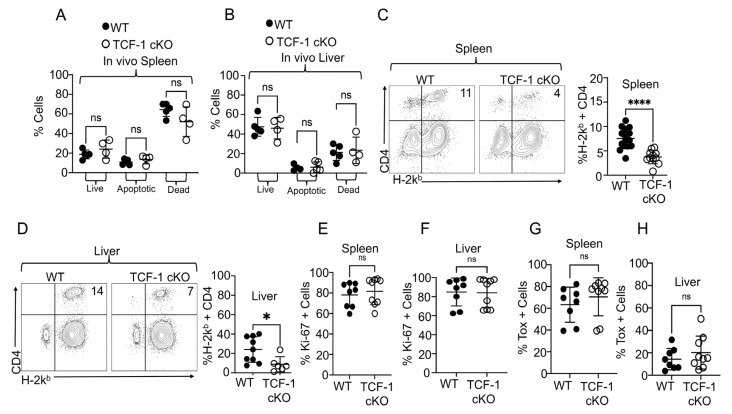
TCF-1 regulates donor T cell presence but has no effect on T cell exhaustion during alloimmunity. Recipient mice were allotransplanted, as described in Figure 1 with CD4 T cells from WT or TCF-1 cKO mice. At day 7, spleens and livers were taken from euthanized recipients. Lymphocytes were isolated and stained for H2Kb to identify donor cells then gated on CD3, CD4, and with annexin V-FITC and LIVE/DEAD near IR. Cells were identified as live (Ann.V-, IR-), apoptotic (Ann.V+, IR-), or dead (Ann.V+, IR+), in both (**A**) spleen and (**B**) liver. Flow plots and statistical analyses of CD4+ H2kb+ donor CD4 T cells from (**C**) spleen and (**D**) liver at day 7 post-transplant. (**E**–**H**) Recipient mice were allotransplanted as before, and at day 7, spleens and livers were obtained from euthanized recipients. Lymphocytes were isolated, and the cells were stained for H2Kb, CD3, CD4, CD8, and Ki67 to identify proliferating/activated cells in (**E**) the spleen and (**F**) liver. Donor CD4 T cells were also stained for TOX to identify the exhausted cells in (**G**) the spleen and (**H**) liver. All individual points are shown with mean and SD. For A-B N = 3–5 per group with one representative of two experiments shown. For C-H, n = 3–5 per group with combined data from two experiments shown. * Means *p*-value ≤ 0.05, **** means *p*-value ≤ 0.0001.

**Figure 7 ijms-24-04326-f007:**
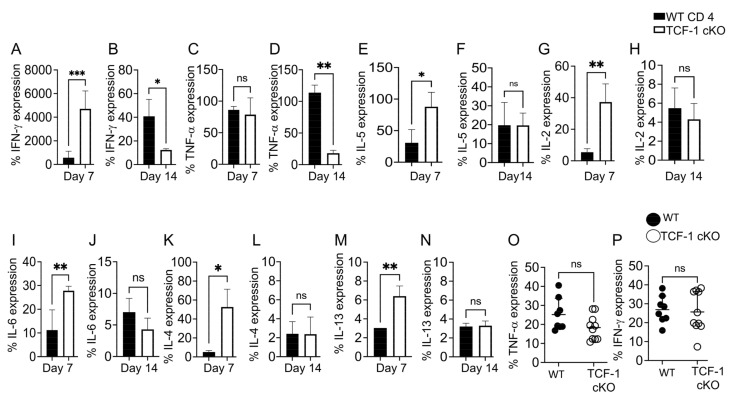
TCF-1 regulates serum levels of cytokines during alloimmunity. Recipient BALB/c mice were allotransplanted with WT or TCF-1 cKO donor CD4 T cells, as before. Serum was obtained from cardiac blood of euthanized recipient mice at day 7 and day 14 post-transplant and was tested using a LEGENDplex multiplex ELISA kit. Serum concentration (pg/mL) over time for WT versus TCF-1 cKO-transplanted mice. (**A**,**B**) day 7 and at day 14 post-transplant of IFN-γ. (**C**) Day 7 TNF-α. (**D**) Day 14 TNF-α. (**E**) Day 7 IL-5, (**F**) Day 14 IL-5. (**G**) Day 7 IL-2. (**H**) Day 14 IL-2. (**I**) Day 7 IL-6. (**J**) Day 14 IL-6. (**K**) Day 7 IL-4. (**L**) Day 14 IL-4. (**M**) Day 7 IL-13. (**N**) Day 7 IL-13. At day 7 post-transplant, donor splenocytes were obtained and restimulated with 6 h of culturing with anti-CD3/anti-CD28 or PBS (control), along with GolgiPlug. Donor cells were stained for H2Kb, CD3, and CD4, then fixed/permeabilized and stained with anti-IFN-γ and anti-TNF-α. Production of (**O**) TNF-α and (**P**) IFN-γ was measured by flow cytometry, using percent cytokine-positive cells. All individual points are shown with mean and SD. n = 3–5 per group for A-K with one representative of two experiments shown. For L-M, n = 3–5 per group with data from two experiments are shown. All data were analyzed with Student’s *t*-test. * Means *p*-value ≤ 0.05, ** means *p*-value ≤ 0.01, and *** means *p*-value ≤ 0.001.

**Figure 8 ijms-24-04326-f008:**
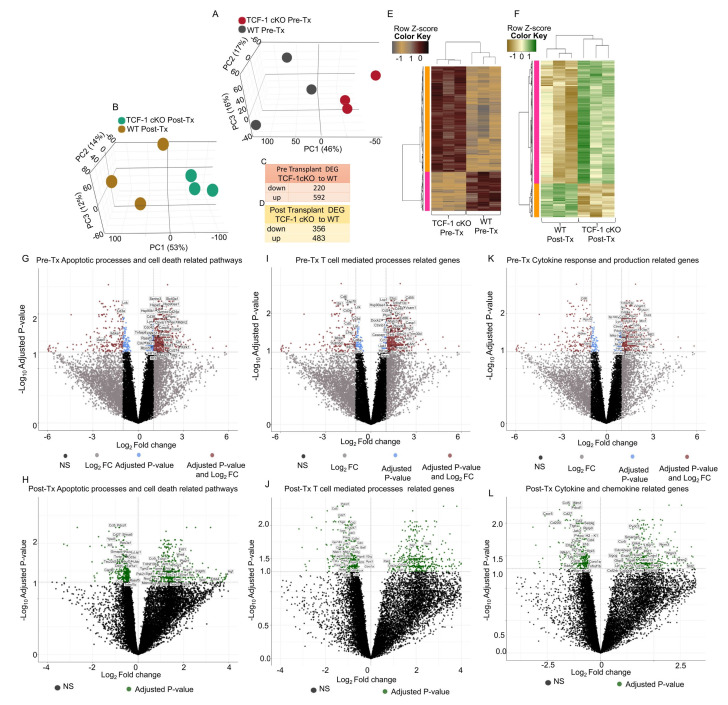
TCF-1 regulates key signaling pathways in donor CD4 T cells. Recipient BALB/c mice were allotransplanted with WT or TCF-1 cKO donor CD4 T cells, as before. Pre-transplant (Pre-Tx) donor CD4 T cells were FACS-sorted from the spleens of donor mice prior to transplant, and at day 7 post-transplant (Post-Tx), donor CD4 T cells were FACS-sorted back from the recipient spleen. Cells were sorted into TRizol, then RNA was extracted, and paired-end sequencing was carried out on an Illumina sequencer. Data were analyzed using the statistical computing environment R (v4.0.4), the Bioconductor suite of packages for R, and Rstudio (v1.4.1106). Three dimensional PCA plots of the pre-transplant samples (**A**) and post-transplant samples (**B**) are shown. Tables showing up- and downregulated differentially expressed genes—DEGs (FDR < 0.05, LogFC = 1) of pre-transplanted samples (**C**), and DEGs (FDR < 0.05) of post-transplanted samples (**D**) are given. (**E**,**F**) Gene ontology enrichment analysis was conducted using the g: Profiler toolset; g: GOSt tool. (**G**,**H**) Volcano plots showing DEGs of pre-transplanted samples (**G**) and post-transplanted samples (**H**), with important genes that play a role in cell death and apoptotic processes are labeled. (**I**,**J**) Volcano plots showing DEGs of pre-transplanted samples (**I**) and post-transplanted samples (**J**), with important genes that play a role in T cell-mediated processes are labeled. (**K**,**L**) Volcano plots showing DEGs of pre-transplanted samples (**K**) and post-transplanted samples (**L**), with important genes that play a role in cytokine and chemokine regulation are labeled. For each condition we used three or four mice.

## Data Availability

No software products, custom code, or algorithms were developed for this manuscript. All resources are available to anyone by request.

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
