# Peer review of "TCF-1 Is Required for CD4 T Cell Persistence Functions during AlloImmunity"

_ijms, 2023, doi:10.3390/ijms24054326_

Round 1

Reviewer 1 Report

Mammadli et al. in their manuscript showed the pathogenic role of TCF-1 in GvHD using conditional TCF-1 knockout mice and control. Although the study does not show fundamental results or novel techniques, it is solid and may be of interest for the readers of the International Journal of Molecular Sciences.

Some minor points should be corrected:

-        Often there is a mix of past and present tense, especially in the abstract

-        Some words require plural, for example mice and cells are plural and require a plural verb

-        In the introduction the results are too detailed and to long explained, please shorten

-        In Fig 1 it is hard to distinguish the different mice strains, please modify

-        The results regarding “TCF-1 regulates CD4 T cell phenotype and memory function” disturb the flow of the Results section, the manuscript may be improved by moving this section to the beginning

-        Line 170: this sentence is unclear to me. Is a gene deletion not always cell intrinsic or what do the authors mean by that?

-        Line 170 – 181: This part may be difficult for the reader to understand and the line “become like TCF-1 cKO mice” in line 173 is not a scientific statement. Please rephrase.

-        Fig 3: How often was this experiment replicated? Could the “technical errors” be avoided. Why was expression of HPRT not detectable, although it is a house-keeping gene? Did the authors measure CD44/CD62L expression in vivo?

-        Fig 6 Why are there different results comparing apoptosis in vitro and in vivo? Should be clarified in the discussion section.

-        Line 422: The authors mention cell exhaustion. Was there an upregulation of exhaustion markers in TCF-1 deficient cells?

-        Fig 2: “Naïve CD4+ T cells from WT…..were euthanized”, I think the authors meant the mice were euthanized, not the cells.

-        Fig 5: Why are differences in apoptosis btw. WT and TCF-1 cKO mice not longer detectable after 72 hours? If the authors state the theory of senescence, it might be unclear to the reader why cKO cells then stop being senescent. Please comment on this.

-        Fig 7: In the figure cytokine levels are depicted as %, but how are this % calculated. The figure legend mentions serum conc in (pg/ml).

-        Fig 8: This figure is hardly readable and should be larger.

-        Why could the authors detect upregulation of apoptotic genes in TCF-1 cko mice in vivo using RNAseq, but not on protein level using Annexin staining?

Author Response

Point by point the reviewers' comments
Reviewer #1: (1) Often there is a mix of past and present tense, especially in the abstract.  Some words require plural, for example mice and cells are plural and require a plural verb

Response to reviewer 1: We thank the reviewer; we have asked a professional editor to help us improve our manuscript.

Reviewer #1: (2) In the introduction the results are too detailed and to long explained, please shorten.

Response to reviewer 1: We have revised the introduction, focused only on our CD4 T cells, and summarized our findings in 2 pages. Thus, we believe that the revised manuscript is significantly improved.

Reviewer #1 (3) In Fig 1 it is hard to distinguish the different mice strains, please modify.

Response to reviewer 1:  We have revised the figure and changed the figure with new colors to Clearfield. (See revised figure 2.

Reviewer #1: (4) The results regarding “TCF-1 regulates CD4 T cell phenotype and memory function” disturb the flow of the Results section, the manuscript may be improved by moving this section to the beginning.

Response to reviewer 1:  Per the reviewer's suggestion, we changed the phenotype to figure 1. Please see page 5, lines 103-181 of the revised manuscript. (See the revised green highlighted)

Reviewer #1: (5) Line 170: this sentence is unclear to me. Is a gene deletion not always cell intrinsic or what do the authors mean by that?

Response to reviewer 1:  We agree with reviewer 1; in the revised paper, revised the sentences.  Please see page 8, lines 177-181 of the revised manuscript. Changes are highlighted.

Reviewer #1: (6) -     Line 170 – 181: This part may be difficult for the reader to understand and the line “become like TCF-1 cKO mice” in line 173 is not a scientific statement. Please rephrase.

Response to reviewer 1:  In the revised manuscriptwe added a paragraph in the revised manuscript. Please see page 8, lines 177-181 of the revised manuscript.

Reviewer #1: (7) Fig 3: How often was this experiment replicated? Could the “technical errors” be avoided. Why was expression of HPRT not detectable, although it is a house-keeping gene? Did the authors measure CD44/CD62L expression in vivo?

Response to reviewer 1:  In the revised manuscriptwe added how section to each figure how often these experiments were conducted. Please see page 37 lines 843, page 38 lines 859, page 39 lines 873, page 40 lines 888, page 41lines 905, page 41 lines 923, page 43 lines 940 and page 43 lines 960. See changes in highlighted.

We did not perform any experiments on in vivo CD44/CD62L. Due to allo activation, most donor T cells are CD44 hi

Reviewer #1: (8) Fig 6 Why are there different results comparing apoptosis in vitro and in vivo? Should be clarified in the discussion section.

Response to reviewer 1:  In the revised manuscript we explain this in the discussion section (See page 23 lines 534-536) page 24 lines 527-540, page 24 line 550 and page 25 lines 560-566.

Reviewer #1: (9) Line 422: The authors mention cell exhaustion. Was there an upregulation of exhaustion markers in TCF-1 deficient cells?

Response to reviewer 1:  In the revised manuscript we added section See page 15 lines 339-342. See highlighted

Reviewer #1: (10) Fig 2: “Naïve CD4+ T cells from WT…..were euthanized”, I think the authors meant the mice were euthanized, not the cells.

Response to reviewer 1:  In the revised manuscript we corrected this mistake section See page 37 lines 827-828.

Reviewer #1: (11) Fig 5: Why are differences in apoptosis btw. WT and TCF-1 cKO mice not longer detectable after 72 hours? If the authors state the theory of senescence, it might be unclear to the reader why cKO cells then stop being senescent. Please comment on this..

Response to reviewer 1:  In the revised manuscript we added a paragraph (See page 23 lines 536-536, page 25 lines 560-566.

Reviewer #1: (12) -        Fig 7: In the figure cytokine levels are depicted as %, but how are this % calculated. The figure legend mentions serum conc in (pg/ml).

Response to reviewer 1:  In the revised manuscript we added a paragraph in method section. (See page 32, lines 723-728, page 33 lines 729-733.

Reviewer #1: (13) Fig 8: This figure is hardly readable and should be larger.

Response to reviewer 1:  In the revised manuscript we enlarge the figure. (See revised figure 8)

Reviewer #1: (14) Why could the authors detect upregulation of apoptotic genes in TCF-1 cko mice in vivo using RNAseq, but not on protein level using Annexin staining?

Response to reviewer 1:  RNA sequences data not always comparable to protein level, thus we perform experiments on transcript level and protein level.

Reviewer 2 Report

In this Mammadli et al., the authors investigate the role of TCF-1 in the disease severity and pathogenesis of gvhd using the B6 to Balbc adoptive transfer model of gvhd. The authors find a significant abrogration of the ability of MHC mismatched T-cells to produce multiorgan inflammation and eventual death in mice. The phenotype is quite striking and pronounced. The authors then performed a series of assays on cells isolated from both the original and adoptively transfered mice to discover the mechanism of why TCF-1 loss results in this phenotype. They discover differences at the level of memory populations, inflammatory cytokines, and transcriptional differences between TCF-1 negative cells and wild type cells.

Although the study provides phenotypic characterization of the role of TCF-1 in the setting of gvhd, further details regarding the explanation of the findings as well as other revisions are required before recommending this manuscript for publication. In particular:

-Consider condensing down the introduction section after “To study the role of TCF-1 in CD4 vells” to 1-2 paragraphs listing the key findings and moving the rest of the info from this section to either the results section or the discussion section. Alternately, some of it can also be deleted to avoid redundancy in the rest of the paper.

-More Specific details regarding the scoring for gvhd should be given in the methods section, including whether or not any subjective parameter was blinded to grouping.

-Were the clinical scores performed also on the BM + CD4+ Cre+/+ mice? They don’t seem to be present on Figure 1C.

-The increase in effector and central memory cells in TCF-1 deficient CD4+ populations is an important finding and may have very well contributed  either entirely or in part to the phenotype observed in this paper as the inability to elicit gvhd by memory T cells has been demonstrated by several groups.  (see Blood 2007 Apr 1; 109(7): 3115-3123) An important experiment to flush this possibility out would be to deplete the memory populations from TCF-1 deficient populations prior to transfer to see if this eliminates the phenotype observed. If not possible, would at least expand upon this possibility in the discussion section.

-The bone marrow chimera studies (Fig 2H-M), although interesting, are relatively difficult to follow and only focused on a small set of phenotypes to determine whether phenotypes are cell intrinsic versus extrinsic. As they do not contribute significantly to the overall paper, would consider eliminate them (or moving entirely to supplemental) to preserve the general flow of the paper.

-The authors conclude their description of the results of Fig5 by saying “These findings highlight that TCF-1 have minimal impact on CD4 T cells exhaustion, proliferation in vivo studies” although the number of CD4 cells in the absence of TCF-1 was lower. A Simple explanation that would unify the in vitro and in vivo results would be that TCF-1 cells are more highly prone to activation induced cell death thus resulting in less cells in the one week after transfer when the in vivo assays were conducted, which would be consistent with the transcriptional data presented later showing increase in apoptotic markers. Would consider explaining the discordant in vitro and in vivo results more or else consider either eliminating the in vitro results altogether or placing in supplementary section.

-For the section labeled “TCF-1 regulates serum levels of cytokines during allo-immunity”, consider condensing down the description of these results, the results of this section can likely be described in 1-2 paragraphs by just listing the cytokines that were increased at day 7 or day 14 rather than individually describing each of them.

-Despite a number of inflammatory cytokines being upregulated at day 7, including IFN-gamma, TNF, and IL-6, this seems to have not resulted in any inflammation and recruitment of additional inflammatory cells such as macrophages at the gvhd sites as indicated by Fig3 (an actually less cell death), which is quite surprising. What is the explanation regarding this? Is it possible that there is a additional immunosuppressive role being contributed by TCF-1 deficient cells that is protecting these sites from inflammation despite being in a pro-inflammatory state at this time point?

Author Response

Reviewer #2 Q1: Although the study provides phenotypic characterization of the role of TCF-1 in the setting of gvhd, further details regarding the explanation of the findings as well as other revisions are required before recommending this manuscript for publication. In particular:

-Consider condensing down the introduction section after “To study the role of TCF-1 in CD4 vells” to 1-2 paragraphs listing the key findings and moving the rest of the info from this section to either the results section or the discussion section. Alternately, some of it can also be deleted to avoid redundancy in the rest of the paper.

Response to reviewer 2:  We agree with the reviewer. In the revised manuscript we shorten the introduction to 2 pages (Please see on page 3, and page 4.

Reviewer #2 Q2: Were the clinical scores performed also on the BM + CD4+ Cre+/+ mice? They don’t seem to be present on Figure 1C.

Response to reviewer 3:  We have revised the figure and changed the figure to figure 2 with new colors.

Reviewer #2 Q3: -The increase in effector and central memory cells in TCF-1 deficient CD4+ populations is an important finding and may have very well contributed  either entirely or in part to the phenotype observed in this paper as the inability to elicit gvhd by memory T cells has been demonstrated by several groups.  (see Blood 2007 Apr 1; 109(7): 3115-3123) An important experiment to flush this possibility out would be to deplete the memory populations from TCF-1 deficient populations prior to transfer to see if this eliminates the phenotype observed. If not possible, would at least expand upon this possibility in the discussion section.

Response to reviewer 2:  We added the suggestion information to the revised manuscript. Please see page 21 lines 487-489 we also added your references to several part of the manuscript.

Also see the revised manuscript page 25 lines 560-566.

Reviewer #2 Q4: -The bone marrow chimera studies (Fig 2H-M), although interesting, are relatively difficult to follow and only focused on a small set of phenotypes to determine whether phenotypes are cell intrinsic versus extrinsic. As they do not contribute significantly to the overall paper, would consider eliminate them (or moving entirely to supplemental) to preserve the general flow of the paper.

Response to reviewer 2:  We agree with the reviewer we have moved the chimera studies to supplementary figure 2. See page 44, lines 980-984, page 45 lines 985-990. 

Reviewer #2 Q5: -The authors conclude their description of the results of Fig5 by saying “These findings highlight that TCF-1 have minimal impact on CD4 T cells exhaustion, proliferation in vivo studies” although the number of CD4 cells in the absence of TCF-1 was lower. A Simple explanation that would unify the in vitro and in vivo results would be that TCF-1 cells are more highly prone to activation induced cell death thus resulting in less cells in the one week after transfer when the in vivo assays were conducted, which would be consistent with the transcriptional data presented later showing increase in apoptotic markers. Would consider explaining the discordant in vitro and in vivo results more or else consider either eliminating the in vitro results altogether or placing in supplementary section.

Response to reviewer 2: We added section explain these result (see page 24 lines 559 page 25 line 560-566.

Reviewer #2 Q6: -For the section labeled “TCF-1 regulates serum levels of cytokines during allo-immunity”, consider condensing down the description of these results, the results of this section can likely be described in 1-2 paragraphs by just listing the cytokines that were increased at day 7 or day 14 rather than individually describing each of them.

Response to reviewer 2: We agree with reviewer, and we shorten the section significantly (See page 16 and 17)

Reviewer #2 Q7: -Despite a number of inflammatory cytokines being upregulated at day 7, including IFN-gamma, TNF, and IL-6, this seems to have not resulted in any inflammation and recruitment of additional inflammatory cells such as macrophages at the gvhd sites as indicated by Fig3 (an actually less cell death), which is quite surprising. What is the explanation regarding this? Is it possible that there is a additional immunosuppressive role being contributed by TCF-1 deficient cells that is protecting these sites from inflammation despite being in a pro-inflammatory state at this time point?

Response to reviewer 2: We agree with reviewer, and we added a section explaining this section (See page 23 lines 535-536, page 24 lines 537-540.

Round 2

Reviewer 1 Report

I think the authors performed sufficient changes to improve the quality of the manuscript for publication.

Reviewer 2 Report

This is a revision of a previously submitted manuscript by Mammaldi et al. The authors have addressed the concerns brought up previously. Although there are still a lot of open questions regarding the mechanistic role of TCF-1 in gvhd, this manuscript provides a nice phenotypic characterization of TCF-1 to build upon for future mechanistic studies.